# Diagnostic performance of molecular and serological tests of SARS-CoV-2 on well-characterised specimens from COVID-19 individuals: The EDCTP "PERFECT-study" protocol (RIA2020EF-3000)

Joseph Fokam [1,2,3,4]*, Claudia Alteri[5,6], Luna Colagrossi[7], Anne-Marie Genevieve[8], Désiré Takou[1], Alexis Ndjolo[1,4], Vittorio Colizzi[1,9,10], Nicaise Ndembi[11], Carlo-Federico Perno[1,7]

1 Chantal BIYA International Reference Centre for Research on HIV/AIDS Prevention and Management (CIRCB), Yaoundé, Cameroon, 2 Faculty of Health Sciences (FHS), University of Buea, Buea, Cameroon, 3 National Public Health Emergency Operations Coordination Centre (NPHEOCC), Yaounde, Cameroon, 4 Faculty of Medicine and Biomedical Sciences (FMBS), University of Yaounde I, Yaounde, Cameroon, 5 University of Milan, Milan, Italy, 6 AVIRALIA Foundation Onlus, Rome, Italy, 7 Bambino Gesu Children's Hospital, Rome, Italy, 8 Association de Recherche en Virologie et Dermatologie (ARVD), Paris, France, 9 University of Rome Tor Vergata, Rome, Italy, 10 Evangelical University of Cameroon, Bandjoun, Cameroon, 11 Africa Centres for Disease Control and Prevention, Abbis Ababa, Ethiopia

☯ These authors contributed equally to this work.
* josephfokam@gmail.com

**Funding:** JF EDCTP RIA2020 EF-3000 European and Developing Countries Clinical Trial Partnership

## Abstract

### Background

The SARS-CoV-2 pandemic is a global threat affecting 210 countries, with 2,177,469 confirmed cases and 6.67% case fatality rate as of April 16, 2020. In Africa, 17,243 cases have been confirmed, but many remain undiagnosed due to limited laboratory-capacity, suboptimal performance of used molecular-assays (~30% false negative, Yu *et al.* and Zhao *et al.*, 2020) and limited WHO-recommended rapid-tests.

### Objectives

We aim to implement measures to minimize risks for COVID-19 in Cameroon, putting together multidisciplinary highly-experienced virologists, immunologists, bioinformaticians and clinicians, to achieve the following objectives: (a) to integrate/improve available-infrastructure, methodologies, and expertise on COVID-19. For this purpose, we will create a platform enabling researchers/clinicians to better integrate and translate evidence into the COVID-19 clinical-practice; (b) to enhance capacities in Cameroon for screening/detecting individuals with high-risks of COVID-19, by setting-up effective core-facilities on-site; (c) to validate point-of-care SARS-CoV-2 molecular assays allowing same-day result delivery, thus permitting timely diagnosis, treatment, and retention in care of COVID-19 patients; (d) to implement SARS-CoV-2 diagnosis with innovative/highly sensitive ddPCR-based assays

https://www.edctpgrants.org/ The funders had and will not have a role in study design, data collection and analysis, decision to publish, or preparation of the manuscript.

**Competing interests:** The authors have declared that no competing interests exist.

and viral genetic characterization; (e) to validate serological assays to identify COVID-19-exposed persons and follow-up of convalescents.

## Methods

This is a prospective, observational study conducted among COVID-19 suspects/contacts during 24 months in Cameroon. Following consecutive sampling of 1,536 individuals, oro/nasopharyngeal swabs and sera will be collected. Well characterised biorepositories will be established locally; molecular testing will be performed on conventional real-time qPCR, point-of-care GeneXpert, antigen-tests and digital droplet PCR (ddPCR); SARS-CoV2 amplicons will be sequenced; serological testing will be performed using ELISA, and anti-body-based kits. Sensitivity, specificity, positive- and negative-predictive values will be evaluated.

## Expected outcomes

These efforts will contribute in creating the technical and clinical environment to facilitate earlier detection of Sars-CoV-2 in Africa in general and in Cameroon in particular. Specifically, the goals will be: (a) to implement technology transfer for capacity-building on conventional and point-of-care molecular assays, achieving a desirable performance for clinical diagnosis of SARS-CoV2; (b) to integrate/improve the available infrastructure, methodologies, and expertise on Sars-CoV2 detection; (c) to improve the turn-around-time for diagnosing COVID-19 infection with obvious advantage for patients/clinical management thanks to low-cost assays, thus permitting timely treatment and retention in care; (d) to assess the epidemiology of COVID-19 and circulating-variants in Cameroon as compared to strains found in other countries.

## Introduction

### Background and rationale

**The unprecedented spread of coronavirus worldwide and in sub-Saharan Africa in particular, calls for an urgent and coordinated continental response.** Three and half months after notification of the first cases of pneumonia related to coronavirus disease in December 2019 (COVID-19) in Wuhan, 210 countries have been affected worldwide for a total of 2,177,469 confirmed cases, with about 19% morbidity (12% mere and 5% severe), a case fatality rate (CFR) of 6.67% (145,304) and a poor recovery of 25.11% (546,475) as of April 16, 2020 [1, 2]. The number of confirmed cases and CFR range respectively from 748,256 and 4.77% (35,688) in the Americas to 17,243 and 5.28% (911) in Africa, with a declining CRF over time [2]. However, the level of risk is postulated to be much more higher in sub-Saharan Africa (SSA) owing to the weak quality of healthcare and surveillance systems. Interestingly, SSA was the last continent where infected cases were reported (partly due to limited diagnostic potentials); the numbers of circulating SARS-CoV-2 cases might be under-reported; and consequently the burden of COVID-19 might be largely under-estimated in terms of morbidity/mortality (due to suboptimal health care systems) and socio-economic impairements. Thus, evidence-based decision-making are needed for timely public health responses [2, 3].

Though Africa currently appears with the lowest number of COVID-19 confirmed cases, the outbreak is propagating very fast throughout the continent [3]. African males are more

affected than women (3/2 M/F ratio), median age is 41 (min-max: 0–88) years, and older males are disproportionately affected [3]. As one of the African countries with high incidence of COVID-19 in Central-Africa, Cameroon has in 2020 reported 1,016 cases out of 4,982 tested [4]. This therefore suggests the need of a coordinated response strategy from country-level to other SSA countries [3, 4], and calls for a large scale and early SARS-CoV-2 detection approach in order to mitigate COVID-19 associated-impairements within the SSA settings.

**Molecular diagnostics of SARS-CoV-2 deserve adapted strategies to limit false reporting.** The viral nucleic acid real-time reverse transcription-polymerase chain reaction (rRT-PCR) assay has become the current standard for the molecular diagnosis of COVID-19. However, there are limitations in the global supply, technical challenges in testing thousands of samples daily, high demands for the rRT-PCR primers, lengthly turnaround times, and varying sensitivities of invitro diagnostic assays. Due to these limitations, the field implementation of rRT-PCR remained suboptimal regarding the need for rapid and simple screening of COVID-19 infected individuals [5].

Regarding rRT-PCR detection of SARS-CoV-2 in different types of clinical specimens, bronchoalveolar lavage fluids yielded the highest positivity (93%), followed by sputum (72%), nasal swabs (63%), fibrobronchoscope brush biopsy (46%), pharyngeal swabs (32%), feces (29%), blood (1%) and urine (0%). Moreover, except for the nasal swab specimen reported with a mean PCR cycle-threshold value of 24.3 ($1.4 \times 10^6$ copies/mL), the mean cycle-threshold values of all other specimen types were greater than 30 ($<2.6 \times 10^4$ copies/mL), suggesting varying viral shedding according to specimens and the need for combining specimen types for improved diagnostic sensitivity [6]. Using an N-gene specific quantitative RT-PCR assay, viral loads in throat swab and sputum peaked around 5–6 days after symptoms onset, ranging from 104 to 107 copies per mL [7, 8]. In contrast, Xi *et al.* showed a peak of infectiousness at 0–2 days before symptom onset [9]. These observations suggest discrepancy in understanding the optimal timing for SARS-CoV2 testing and the need for further investigations [8, 9]. Of note, median duration of SARS-CoV-2 RNA shedding appears around 12, 18 and 19 days after symptoms onset in nasopharyngeal swabs, stools and sputum respectively [10–12]. Regarding disease severity, the mean viremia of severe cases was significantly higher than that of mild cases, suggesting that higher viremia is a predictor of severe/critical outcomes [10, 13]. This is likely true as 90% of mild cases become negative on RT-PCR by day 10 post-onset of symptoms while most severe/critical cases remain positive after 10 days of COVID-19 post-symptom onset [11]. The U.S FDA granted Emergency Use Authorization (EUA) for PCR equipment for SARS-CoV-2 testing, of which: (1) Da AN Gen SARS-CoV-2 (Sun Yat-sen University); (2) Abbott Real Time SARS-COV-2 (Abbott Molecular); (3) Xpert Xpress SARS-COV-2 Test (Cepheid), and (4) Panther Fusion SARS-COV-2 (Hologic Inc.). Among these assays, the Xpert revealed satisfactory positive (99.5%) and negative (95.8%) agreements for SARS-CoV-2 detection in a variety of upper- and lower-respiratory-tract specimens. Its high sensitivity, short time-around-time (45 min) and simple procedure even for non-laboratorians, the Xpert would be helpful in case of clinical emergency [14]. In this context, deploying such device through mobile laboratories equipped with appropriate biocontainment may contribute substantially in the pandemic control (i.e. active case finding of tuberculosis among high-risk targets in South Africa) [15]. Thus, applying these concepts in SSA will be of great relevance in defining testing strategies [10, 16].

A major drawback of rRT-PCR for the detection of SARS-CoV-2 remains its suboptimal sensitivity and the absence of an absolute quantitative evaluation of SARS-CoV-2 viral load [12, 16]. By using digital droplet PCR (ddPCR) as gold standard (lower detection limit of 10 copies) to evaluate the performance of a rRT-PCR set at 38 cycle threshold (Ct) for positivity, the sensitivity reported by Yu *et al.* was substandard (~68%) [12] and similar to sensitivity

reported by Zhao *et al.* (~67%) [16]. Interestingly, the Ct of qRT-PCR appears to be highly correlated with the copy number of ddPCR for both the ORF1ab (R2 = 0.83) and the nucleocapsid (R2 = 0.87) genes, while only 61.2% of single-gene positive samples on qRT-PCR were also detected on ddPCR (viremia ranging from 11.1–123.2 copies) [12]. In the frame, we also applied an in-house ddPCR assay targeting RdRp for SARS-CoV-2 detection on negative swabs after RTPCR. This developed assay accurately identified and quantified SARS-CoV-2 even at 2.9 copies per reaction (i.e. 28 viral copies/mL), thus suggesting an excellent analytical sensitivity. Moreover, as quantification per mL is highly dependent on the quality of sampling and extraction procedure, this assay includes an internal reference gene (RNAseP) for quality control. The presence of this reference gene excludes any eventual PCR inhibition and confirms successful RNA extraction, thus reducing substantially the risk of false negative results. Moreover, ddPCR will allow a linear quantitation of SARS-CoV-2 viral load, and the determination of viral load in clinical samples will henceforth supports an indepth interpretation of laboratory assays, which in turn will strengthen the strategy for case isolation, tracking and contact tracing. It is becoming common to assume that the Ct values from RT-PCR diagnostic tests are direct estimates of viral load and Ct values could serve in identifying non-infectious patients in spite of positive PCR or infectious samples in correlation to cell cultures. Even though using the Ct values as estimates of SARS-CoV-2 viral loads appears simple and routinely acceptable, it could be misleading and the risk of introducing errors might be consistent. Thus, using ddPCR at a second diagnosis level could help in an accurate quantitation of SARS-CoV-2 viral load and an appropriate management of SARS-CoV-2 positivity [12].

**Validation of existing SARS-CoV-2 antigen and serological assays could enhance case-finding and surveillance of COVID-19 in sub-Saharan Africa.** The limited laboratory capacity for SARS-CoV-2 molecular testing and the global stock-outs of reagents raised interest on using rapid and/or easy-to-use devices to facilitate screening in SSA, based either on SARS-CoV-2 antigen detection in sputum/throat/swab samples, or human antibodies detection in blood/sera/plasma specimens [17, 18]. However, the performance of these rapid assays remains largely unknown within SSA populations. Rapid antigen tests are designed to detect the presence of a viral antigen (targeting usually the Nucleocapsid), and the tests are inexpensive, easy to use, have short turn-around time, suitable for point-of-care (PoC), but with a lower diagnostic sensitivity (provide reliable results only with high viral-loads ≥100,000 copies/mL) especially during the first days after symptoms onset [19–21]. Due to this limitation, WHO, CDC and ECDC recommend the use of rapid antigen tests with a subsequent confirmation of negative results by RT-qPCR [22–24]. Among the 3rd generation antigen-detecting tests LumiraDx, Panbio COVID-19 Abbott Ag Rapid Test and Standard Q COVID-19 Ag Test Biosensor comply with the minimum performance requirements of the WHO [23].

Regarding serological assays, performance varies according to manufacturers (https://www.fda.gov/medical-devices/coronavirus-disease-2019-covid-19-emergency-use-authorizations-medical-devices/eua-authorized-serology-testperformance); FDA recommended antibody tests with highest performance are Abbott Architect SARS-CoV-2 IgG (100% sensitivity, 99.6% specificity), Abbott Laboratories Advise Dx SARS-CoV-2 IgM (95.0% sensitivity, 99.6% specificity), Assure Tech. Assure COVID-19 IgG/IgM Rapid Test Device (100% sensitivity for IgM and 90.0% for IgG; 98.8% specificity for IgM and 100% for IgG) and the EUROIMMUN SARS-COV-2 ELISA IgG (90.0% sensitivity and 100% specificity). Among them, Abbott ARCHITECT SARS-CoV-2 IgG assay showed a satisfactory performance, with excellent sensitivity and specificity after 14 days of sympthom onset [25–27]. The availability of neutralizing antibody detection kit, such as the approved GenScript cPass SARS-CoV-2 Neutralization Antibody Detection Kit (100% sensitivity and specificity) allows the discrimination of neutralizing antibodies at population level [26, 27].

Validation of these SARS-CoV-2 RDTs in SSA populations/settings is of paramount importance for optimal clinical use and positioning. On one hand, antigens are expressed only when the virus is actively replicating (greater performance during acute/early infection), while false-positive results could occur if strips recognize viral antigens from other human-coronaviruses [17]. Thus, validating the use of antigen RDT would reduce events of false negative-results, ensure rapid identify of cases, and minimise the need for expensive molecular testing [17]. On the other hand, COVID-19 antibody-tests might be convenient mainly during recovery as antibody response often occur in the second week post-infection or immunisation. Consequently, antibody testing might lead to delayed clinical-intervention or transmission-prevention [28, 29]. Moreover, COVID-19 antibody test might have cross-reactivity with other human-flu and/or coronaviruses, and the strength/breath of antibody response might vary with age, nutritional status, disease-severity, therapeutic-interventions, or infections like HIV or immunodeficiency-associated comorbidities [30]. Using an enzyme-linked immunosorbent assay (ELISA), Zhao *et al.* showed high rates of seroconversion based on total antibody (93%), IgM (83%) and IgG (65%), with seroconversion time ranging from 11–14 days [11]; the presence of antibodies rapidly increased to 100% (total antibody), 94% (IgM) and ~80% (IgG) as from 15 days after symptoms onset. These are consistent with evidence from Tan *et al.*, revealing that anti-nucleo-capsid-IgM starts on day-7 and peaks on day-28, while IgG ranges from day-10 to day-49 [11]. Interestingly, combining RNA and Antibody testing might improve the overall sensitivity for detecting COVID-19 case, pending confirmatory findings in real-life situations [11].

With scarcity of data as of April 16, 2020 within the African populations/settings, WHO does not recommend the use of these rapid tests for patient care, but strongly encourages research on the potential diagnostic utility of antigen-detecting rapid tests and the continuation of work to establish the usefulness of COVID-19 antibody testing in case management and in epidemiological surveillance [31]. Such evidence would inform global policy on the use of immunodiagnostic rapid tests, and on specific conditions/settings where such tests can be useful for clinical decision, epidemiologic understanding, and/or infection control without referring to molecular assays [28, 29, 31].

## Research hypotheses

1. Research on COVID-19 diagnostic approaches will integrate and improve the available infrastructure, methodologies, and expertise on COVID-19 in Cameroon and other SSA countries;

2. Evidence on the performance of real-time PCR devices in the clinical diagnosis of SARS-CoV2 in SSA would help in identifying the most efficient platforms and in adapting result interpretation according to cycle threshold (Ct) for confirmation of COVID-19 in SSA settings;

3. Evidence on the performance of COVID-19 rapid diagnostic tests would help in identifying the most reliable assays and in designing the most suitable algorithm for use of rapid tests in clinical management and in epidemiological surveillance of COVID-19 in SSA settings;

4. Evaluating the sensitivity of a combination of nucleic acid and RDT would significantly improve the efficiency in detecting COVID-19, both in the even and late phases of infection.

## Study objectives

**Main objective.** With the goal to support public health measures for prevention and control of COVID-19 infections in Africa, our study aims at evaluating the diagnostic

performance of molecular tests and serological rapid kits for the detection of SARS-CoV2 among individuals tested for COVID-19 in the Cameroonian context.

## Specific objectives

1. **To integrate and improve the available infrastructure, methodologies, and expertise on COVID-19**. For this purpose, we will create a platform enabling researchers and clinicians to better integrate and coordinate their research and translate the project outcomes into scientific and clinical expertise for the management of COVID-19 infection within the African context;

2. **To enhance the capacities of the involved centers in Cameroon to screen and detect individuals at risk for COVID- 19 infection, by putting in place effective screening mechanisms and facilities at core sites (based on molecular and serological detection methods and SARS-CoV-2 genome sequencing)**. In this frame, individuals will be considered at risk of COVID-19 infection according to surveillance case definition for human infection with novel coronavirus published by WHO [WHO/COVID-19/Surveillance/v2020.2, January 15 2020];

3. **To determine the performance of sensitive and specific point-of-care molecular detection assay for SARS-CoV-2**. Previous data on Zaire Ebola virus suggested that the POC molecular test has comparable diagnostic accuracy to the conventional real-time, minimized the time to obtain a result, thereby allowing clinicians or public health staff to make expeditious decisions. In the COVID-19 case, nasopharyngeal and oropharyngeal swab, and saliva will be valid samples to obtain a rapid and non-invasive screening for all travelers arriving in Cameroon. Concordance with other molecular tests already available for COVID-19 will be assessed (Corman et al. 2020; Chu et al. 2020);

4. **To apply a rapid and sensitive ddPCR-assay SARS-CoV-2 quantification method as complementary to the standard RTPCR for COVID-19 diagnosis**. This test (single-step ddPCR assay) will have the ability to detect SARS-CoV-2 genome at very low copies number, representing a chance to detect infected subjects before any symptom onset, and to reveal the presence of COVID-19 in the environment in close contact with infected individuals. Based on ddPCR, a linear evaluation of SARS-CoV-2 load will be also available;

5. **To validate a large panel of serological assays to identify COVID-19-exposed or vaccinated persons and to follow-up of COVID- 19 convalescents or immunisation**. This panel will enable an appraisal of Nucleocapside (N) and Spike protein (S) used in screening for natural and artificial immunity respectively;

6. **To evaluate the variability of circulating SARS-CoV-2 variants in Cameroon based on whole genome sequencing and phylogenetic approaches**. Through sequencing of SARS-CoV-2, evidence on the dynamics of COVID-19 from the lineage of origin to newly emerging variants will be delineated, with the goal in predicting the trends of SARS-CoV-2.

## Methods

### Study design

**Study type and population**: Prospective, cross-sectional, observational study among individuals in Cameroon;

**Study sites**: The Chantal BIYA International Reference Centre (CIRCB) is the reference study site for the project, in connexion with the routine national laboratories network for COVID-19 testing in the country [4], as per the national guidelines (Fig 1).

**Description of the Cameroonian settings.** In response to COVID-19, Cameroon has developed a national guideline for COVID-19 prevention and control in the country. This guideline entails case definition, case notification, clinical diagnosis of COVID-19, laboratory diagnosis of SARS-CoV2, case management of COVID-19, surveillance strategies at entry points and at the health facilities, triage approaches, etc. Based national response, 4,982 samples have been tested from suspects for 1,016 (20.39%) confirmed SARS-CoV2 positive by real-time qRT-PCR. Majority is from the political capital region of Yaoundé (558), followed by the economical capital region of Douala (381); male/female sex ratio is 3/2 with a median of 39 (min-max: 0–81) years old; the case fatality rate (CFR) is 2.00% (22) for 16.27% recovery rate (https://www.worldometers.info/coronavirus/). Apart from travellers for whom a conventional real-time polymerase chain reaction (rt-PCR) test is required, testing for COVID-19 is performed by using the antigen rapid detection test (RDT) as the screening tool, and eventually followed by a PCR if negative antigen RDT in a person with suspicion of COVID-19 symptom or a high-risk contact, as per the national guidelines, as shown in Fig 2 [4].

**Study reference site: Virology Laboratory of CIRCB** (http://www.circb.cm/btc_circb/web/ ) The "Chantal BIYA" International Reference Center for research on HIV/AIDS prevention and management (CIRCB) is a government institute of the Ministry of Public Health established on February 2006, of conducting research and clinical monitoring of HIV and co-infections/comorbidities, with compliance to QC/QA and proficiency testing. CIRCB is a WHO-Candidate as National HIV Drug Resistance Reference laboratory, providing technical support for viral-sequencing (PHIA survey, etc) and training of health-professionals. CIRCB has a biobank dedicated for the storage of well characterised infectious samples collected from heath facilities and within research projects, in accordance with GCP and ethical regulations. The standard operational procedures (SOP) for the biobanking are available at www.circb-cm/LBIOSOP-English (English version) and www.circb-cm/LBIOSOP-French (French version). Samples collected are plasma, serum, buffy-coat, blood, DNA, RNA, proteins, PBMCs, tissues, smears, saliva/sputum, infectious pathogens (HIV, HBV, HCV, HPV, etc). In response to COVID-19, CIRCB is a reference laboratoriy for SARS-CoV2 testing, a member of the National COVID-19 public health operations centre, a member of the national COVID-19 genomic surveillance plateform, and a member of the scientific taskforce for COVID-19 in the country.

**Study duration**: The study will run for a duration of two years, with an active enrolment phase during the first 6 months of the project, and subsequent laboratory analyses.

**Sampling method**: A consecutive non-randomised sampling strategy will be used for participants enrollment;

**Target population**: Individuals visiting our routine sample collection site for COVID-19 screening within the framework of the established laboratory network in place, CIRCB being the study reference laboratory.

## Eligibility criteria

**Inclusion criteria**: Every individual found in Cameroon at the moment of the study, suspected/contact COVID-19 case as defined by the WHO, aged 21 years and above, and provide a written informed consent for participation;

**Non-inclusion criteria**: Suspected case without clinical data, any suspected case managed out of the country guidelines, or any suspected case without blood/respiratory fluid specimen available;

REPUBLIQUE DU CAMEROUN
Paix - Travail - Patrie

REPUBLIC OF CAMEROON
Peace – Work - Fatherland

MINISTERE DE LA SANTE PUBLIQUE
---------------
SECRETARIAT GENERAL
--------------------

MINISTRY OF PUBLIC HEALTH
----------------
SECRETARIAT GENERAL
--------------------

Lettre circulaire n° 136-13 MINSANTE/SG/DPML du 17 AVR 2020
relative à la décentralisation du diagnostic moléculaire du Covid-19

LE MINISTRE DE LA SANTE PUBLIQUE
A

- Mesdames et Messieurs les Délégués Régionaux de la Santé Publique;
- Mesdames et Messieurs les Responsables des Formations Sanitaires.

Dans le cadre de la poursuite du processus de décentralisation du diagnostic moléculaire du Covid-19 déjà mis en œuvre au Centre Pasteur du Cameroun de Yaoundé, à l'Hôpital Laquintinie de Douala, et au Centre Pasteur de Garoua, les laboratoires dont les noms suivent sont retenus pour ledit diagnostic.

Il s'agit de :

1.  Région du Centre :
    ➤ Centre International de Référence Chantal Biya (CIRCB)
    ➤ Centre de Recherche en Santé des Armées (CRESAR)
    ➤ Laboratoire National de Santé Publique (LNSP)
    ➤ Centre des Biotechnologies, Nkolbisson, UY1
    ➤ Laboratoire du Centre de Recherche sur les Maladies Emergentes et Réémergentes (CREMER) , MINRESI.
2.  Région de l'Est :
    ➤ Laboratoire de l'Hôpital régional de Bertoua.
3.  Région du Littoral :
    ➤ Laboratoire du Centre de Pneumophtisiologie (Référence pour la prise en charge de la Tuberculose) (Douala).
4.  Région Nord-Ouest :
    ➤ Laboratoire de Référence de la Tuberculose , Hôpital Régional (Bamenda).
5.  Région Ouest :
    ➤ Laboratoire DREAM (Dschang).
6.  Région Sud-Ouest :
    ➤ Laboratoire des maladies infectieuses émergentes, Université de Buea;
    ➤ Laboratoire CBC Health Services Complex, Mutenguene.
7.  Région de l'Adamaoua :
    ➤ Laboratoire de l'Hôpital Régional de Ngaoundéré.
8.  Région Extrême Nord :
    ➤ Laboratoire de l'Hôpital Régional de Maroua.

L'enrôlement desdits laboratoires se fera de manière progressive en fonction de la disponibilité en réactifs et du type de plateforme utilisée sous la coordination du Centre Pasteur du Cameroun, le Laboratoire National de Santé Publique et le Centre de Recherche en Santé des Armées.

AMPLIATIONS :
-   MINSANTE/CAB
-   SESP/CAB
-   SG
-   IGSPL
-   DPML
-   Intéressés/Archives/Chronos-

Site web : www.minsante.gov.cm
E-mail : dpml.cmr@gmail.com

LE MINISTRE
The Minister

Dr. Manaouda Malachie

Tel : (237) 222219281

**Fig 1. Network for COVID-19 testing laboratories.** COVID-19: Coronavirus Disease 2019.

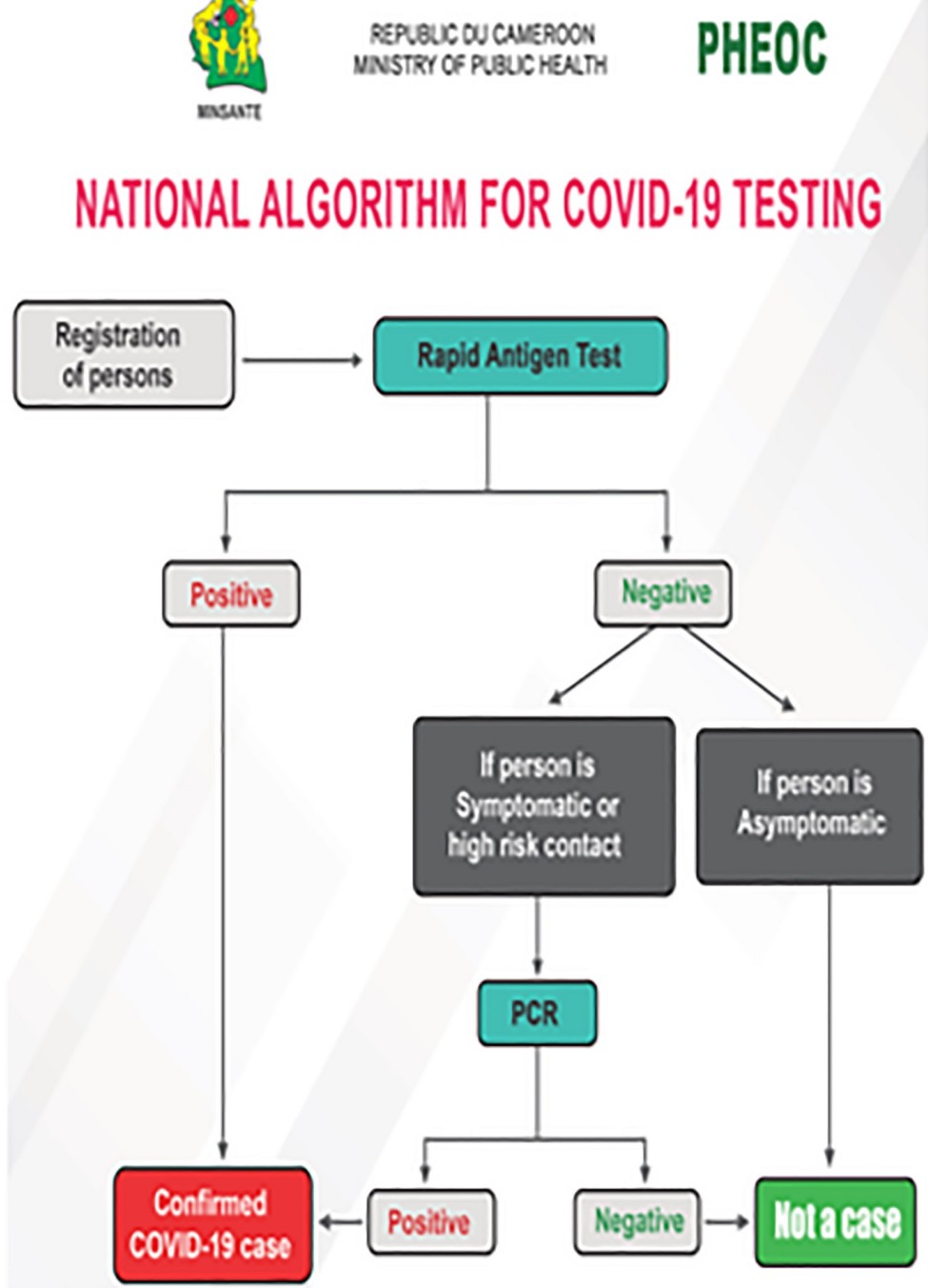

**Fig 2. Standard algortihm for routine COVID-19 testing in Cameroon.** COVID-19: Coronavirus Disease 2019.

**Exclusion criteria**: Enrolled participant with neither results of real-time PCR, serological testing, nor SARS-CoV2 sequence data following positive-PCR for the respective analysis.

## Sample size

The minimum sample size required for the study was calculated based on the positivity rate of SARS-CoV2 among COVID-19 suscpected/contact cases in the country [4], using the following formula: N = z2 x p x (1-p) / e2 With p being the proportion of positive SARS-CoV2 among suscpected cases (p = 51%); with a confidence interval of 95% (z = 1.96); and e being the sampling error rate set at 5%. N = 1.962 x 0.51 (1–0.49) / 0.052 = 384.0. Thus, the sample size is rounded up to 1,536 participants for statistical significance.

## Study procedures and required materials

**Enrolment of study participants and sample collection.** Individuals seeking for COVID-19 testing at the reference sample collection sites will be provided with an information notice on the present study; detailed explanation on the study with be done to each of the potential participant, and then a written informed consent will be signed prior to enrollment. Eligibility criteria will then be verified as aforementioned for consideration as study participants. Nasopharyngeal/oropharyngeal swabs and blood/serum specimens will be collected for each eligible participant. Swabs will be stored in the viral transport media (for real time PCR) while blood/serum will be stored in test tubes (for serology/rapid testing respectively). Collected samples will be kept on cold chain and transported to the reference laboratory unit by well-trained personnel.

**Establishment of well-characterised respiratory and sera specimens of COVID-19 positive cases.** Training of staffs on sample collection procedure, transportation, storage and monitoring conditions will be done as previously described by Wang et al., 2020 [6]. Specimens of oropharyngeal and nasopharyngeal specimens, whole blood/serum, will be collected by trained staffs of the CIRCB in Cameroon.

**Serological and antigen testing of COVID-19 suspected cases: Antibody-based testing.** For now, there are about five antigen-detection RDTs and 27 antibody-detection RDTs that have been selected for the first round of evaluation worldwide (and more are in evaluation), following approvals by FDA and EUA, and consideration for assessment by the Foundation for Innovative New Diagnostics (FIND), available at https://www.finddx.org/covid-19/sarscov2-eval-immuno/.

Based on preliminary results on 100 samples, five most efficient commercially available serological tests (Elisa tests: Dia.Pro COVID-19 IgG/M + confirmation and Aliphax WANTAI SARS-CoV-2; CLIA: Diasorin CLIA LIASON SARS-CoV2SI/S2 IgG; Rapid: Aliphax Cellex q Rapid Test; VivaDiag COVID-19 IgM/IgG rapid Test) were selected, with sensitivity and specificity ranging between 75%-98%. Antigen detection tests to be evaluated are LumiraDx Ag, Panbio COVID-19 Abbott Ag Rapid Test and Standard Q COVID-19 Ag Test Biosensor. Based on local neeeds, INDICAID AgRDTs (Phase Scientific), Ninonasal AgRDTs (NG. BIO-TECH Laboratories) and NG.TEST/IgG-IgM COVID-19 (NG. BIOTECH Laboratories) could be considered.

**Molecular testing of COVID-19.** Molecular testing entails conventional real-time PCR, point-of-care GeneXpert Express (Cepheid), ddPCR as well as sequencing reactions. For all molecular testing and sequencing, viral RNA was extracted from respiratory samples using the QIAmp viral RNA Mini kit (Qiagen Hilden, Germany) according to the manufacturer's protocol.

*Real-time qRT-PCR for SARS-CoV2 testing.* This protocol is designed to detect SARS-CoV2 in human clinical specimens. The two monoplex assays are reactive with the 2019 novel coronavirus, considering: (1) the genetic diversity of 2019-nCoV in humans and animals, (2) the use of an internal control, (3) the kit being widely available across SSA. Amplification will be performed on Applied Biosystem/Thermocycler devices (TheromFisher, ViiATM 7 Real-Time PCR), with a positive control, primer sets, primer and probe sequences. The amplicon sizes of Assay 1 and Assay 2 are 132 bp and 110 bp, respectively. Interpretation of positivity of N gene RT-PCR and Orf1ab is done at required cycle thresholds (Sun Yat-sen University protocol). A negative N gene positive/Orf1b refers to the absence of SARS-CoV2. Results will be evaluated with reference to ddPCR.

*GeneXpert for SARS-CoV2 testing*: *Xpert® Xpress SARS-CoV-2.* Xpert® Xpress SARS-CoV-2 is a test that has the Food and Drug Administration (FDA) emergency use authorization (EUA). Xpert® Xpress SARS-CoV-2 is currently the most prominent test for real-time RT-PCR rapid test for SARS- CoV-2 developed by Cepheid, which provides results in just 45 min using GenXpert benchtop system. It is a rapid and automated point-of-care (POC) molecular test that enables the qualitative detection of SARS-CoV-2 in nasopharyngeal swab, nasal wash, or aspirate specimens from COVID-19 suspects. The test requires only a minute for sample preparation, employs cartridges, and targets multiple regions of the viral genome. Accuracy will be evaluated using ddPCR as gold standard.

*Use of ddPCR SARS-CoV-2.* Droplet digital PCR (ddPCR, Bio-Rad) is an innovative low-cost molecular assay delivering results within 2 hours, for timely clinical decision. ddPCR is based on the principles of limited dilution, end-point PCR, and Poisson statistics, with absolute quantification [32–36]. The single-step ddPCR detects COVID-19 genome at very low copies number, representing a chance to detect infected subjects before symptom onset, including their close contacts. ddPCR system will use a set of primers and probe sets, in order to ensure viral detection at early stages, for timely management. The developed ddPCR will serve for the linear quantification not only of viral genome but also intermediate of viral replication such as subgenomic RNAs. Negativity of these intracellular markers will serve as surrogate of viral clearance, a crucial aspect in breaking the transmission chain. This type of second level diagnosis requires a practical training of staff, with hands-on, in collaboration with expertise in molecular virology within the network in place.

**Illumina next generation sequencing.** Through Next Generation Sequencing (NGS), the entire SARS-CoV-2 genome will be characterized. The use of an NGS Core (equipped with MiSeq, NextSeq500 and NovaSeq 6000 and Bioinformatics Core) will be set up to determine the variability of the entire sequence of SARS-CoV-2 genome in patients' samples. This approach will ensure a full analysis of all intra-host viral variants (even those with low prevalence), thanks to its high-throughput performance and to define the evolution of SARS-CoV-2 epidemic in SSA.

Sequencing of the entire viral genome will be performed in a subset of 50% (n = 384) selected samples nasopharyngeal swabs with multiplex PCR and sequencing (1,000x coverage). Viral genome sequences will be obtained by NGS and analysed along with sequences collected from GISAID database (https://www.gisaid.org/). Similarly, full-length sequences of other human SARS-related coronaviruses and those isolated from animal species, known as natural reservoirs of coronaviruses, will be retrieved from GenBank (http://www.ncbi.nlm.nih.gov/genbank/).

Multiple phylogenetic trees will be constructed by RaxML and MrBayes programs to estimate the evolutionary rates along the entire SARS-CoV-2 genome and to calculate the divergence between SARS-CoV-2 and other related human and animal coronaviruses, by setting best-fit substitution models (nucleotide and amino acids). The ratio of non-synonymous and

synonymous substitutions (dN/dS) will be analysed across the whole sequences of all SARS-CoV-2 coding regions McDonald-Kreitman Test. This analysis will allow to identify viral regions enriched of sites under negative selective pressure, thus less prone to accumulate amino acid mutations. Shannon Entropy (Sn) will be also calculated as additional measure of the extent of amino acid variability at each position of SARS-CoV-2 proteins. Evolutionary divergence of the different genetic viral regions will be also estimated as the extent of nucleo-tide substitutions per site determined by the Tajima-Nei model [37]. These bioinformatics analyses are also geared toward conserved/negatively selected regions of SARS-CoV-2 shared also by the other coronaviruses, for optimal antiviral drug candidates and vaccine targets for pan-coronaviruses.

This study will also take advantage of a large clinical cohort including patients with a well-characterized clinical and virological history for SARS-CoV-2 related disease and with an available SARS-CoV-2 full-length sequence. These patients will be stratified according to the severity of SARS-CoV-2 clinical manifestations (pneumonia vs mild respiratory symptoms). Each nucleotide and amino acid mutation detected in SARS-CoV-2 genome sequences will be analyzed by using an advanced mathematical model developed in collaboration with Yale University. In particular, we will apply a Bayesian variable partition model, and a recursive model selection procedure (according to Zhang et al., PNAS 2010) in order to define mutations either independently or interactively correlated with the severity of SARS-CoV-2 manifestations. Thus, this mathematical model will allow identifying not only single mutation also clusters of mutations in multiple viral genes correlated with cov-2 related severe pneumonia. A multivari-able logistic regression analysis will be also carried out in order to support the correlation of the identified mutations and other clinical and virological parameters with SARS-CoV-2 dis-ease severity. Once more, such second-level diagnosis requires a thorough practical training of staff, in collaboration with experts within the netwrok in place.

**Nanopore point-of-care sequencing.** The cDNA libraries will be quantified using Qubit fluorometer (ThermoFisher Scientific, Waltham, MA) and the sizes of the libraries will be measured using Agilent Bioanalyzer (Agilent Technologies, Santa Clara, CA). For nanopore sequencing in both Cameroon, amplified cDNA libraries from Nextera procedures will be end-repaired and ligated with adapter and motor proteins using 1D library prep kit according to the 1D library preparation protocol for amplicons. Nanopore libraries will be run on Oxford Nanopore flow cells (R9.4 or R9.5) either on a MinION MK1B or a GridION X5 (Oxford Nanopore Technologies, Oxford, UKo). We aim for a target read coverage level of 500–1000 reads per site and majority consensus genome sequences will be generated from the NGS data for target pathogens (e.g., EBOV) for analysis with BEAST. Sequence generated by nanopore will be evaluated for reliability using the Next Generation sequencing (Illumina) as gold standard.

## Ethical considerations

The present study will be conducted according to the declarations of Helsinki on ethical prin-ciples for medical research involving human subjects. (http://www.wma.net/en/30publicat ions/10policies/b3/) Ethical clearance has been obtained from the Cameroon National Ethics Committee for Research on Human Health in Cameroon (http://www.cameroon-ethics.cm/fr/ quisommesnous), under the reference N˚2022/01/1430/CE/CNERSH/SP of January 20, 2022. Participants will provide their signed written informed-consent (see S1 and S2 Files), for ille-rate participants a fingerprinting system will be used alongside a translator whenever neces-sary; phlebotomy will be done by well trained personnel (venipuncture, oro/nasopharyngeal swabs) and performed by trained-staffs Privacy and confidentiality will be ensured, through

the use of unique identifiers and a protected database. Laboratory results will be free of charge for clinical management whenever necessary. A data safety and monitoring will ensure for an appropriate monitoring of the study implementation, under the scientific advisory board of the CIRCB.

**Statistical analysis.** Data will be collected on standard tools (S3 File) by trained field epidemiologists and stored in a password-protected database to ensure confidentiality of participant data and their privacy at the local institution in Yaoundé-Cameroon, by experienced statisticians who will coordinate data management and analyses using SPSS software, according to Good Clinical Data Management Processes (GCDMP).

Diagnostic Performance will be evaluated regarding the sensitivity, specificity, positive/negative predictive values (with 95% Confidence Intervals) will be described using ddPCR as molecular gold standard assay (lower detection limit of 10 copies/test) and ELISA as serology gold standard, stratified by range of viral loads, Ct values, duration after symptom onset, target biomarkers and viral diversity. To provide the highest level of evidence, adjustments of confounders (age, gender, HIV status, onset of symptoms, site, etc) will be performed using logistic and linear regression, wherever applicable.

## Expected outcomes

Our findings will lead to the following primary and secondary outcomes are the sensitivity, specificity, overall agreement, positive and negative predictive values of conventional real-time RT-PCR platforms and point-of-care devices for the detection of SARS-CoV2; the threshold values of real-time PCR assays for an adapted interpretation of SARS-CoV2 results; the sensitivity, specificity, overall agreement, positive and negative predictive values of antigen and antibody rapid tests for the detection and epidemiological surveillance of SARS-CoV2; and the main viral strains of SARS-CoV2 circulating within the Central Africa setting among confirmed COVID-19 cases. Specifically:

- **The specific objective** 1 will provide on the evidence on the performance of conventional real-time qRT-PCR platforms and Xpert Xpress SARS-COV-2 Test (Cepheid), when compared to ddPCR system results, for use as support to the global efforts in controlling COVID-19 pandemic in SSA; evidence on correlations between cycle thresholds obtained by conventional real-time qRT-PCR platforms and Xpert Xpress SARS-COV-2 Test with SARS-CoV-2 load by absolute ddPCR quantification for adapting the interpretation of SARS-CoV-2 results in sub-Saharan African countries; and evidence on available point-of-care molecular assays achieving a desirable performance for the detection of SARS-CoV-2 in resource-limited settings like sub-Saharan Africa.

- **The specific objective 2** will provide evidence of antigen-based assays (like LumiraDx, Panbio COVID-19 Abbott Ag Rapid Test and Standard Q COVID-19 Ag Test Biosensor) performance to allow for accurate and rapid detection of early/acute COVID-19 cases; evidence of antibody-based assays for a timeous SARS-CoV-2 sero-surveillance in SSA; and types of antibody-based assays appropriate for assessing exposure and/or immune response following SARS-CoV-2 infections.

- **The specific objective 3** will generate knowledge on the circulating strains of SARS-CoV-2 in West and Central Africa, and their potential differences with SARS-CoV-2 strains found in other settings; effects of the SARS-CoV-2 genetic diversity on vaccine development at global level, and/or at continental level; phylogeography and phylodynamic of SARS-CoV-2 strains for understanding the migration of the virus toward effective preventive interventions, identification of clusters of transmission and optimal surveillance; and understanding

of the clinical significance of SARS-CoV-2 genetic diversity to rate of recovery or risk of severity and mortality among COVID-19 confirmed cases.

- **The specific objectives 4–6** will provide timely inform policy-makers on SARS-CoV-2 confirmation molecular assays for decision-implementation within the national territory, and possible in other West and Central African countries; proposition of a potential serial or parallel algorithms for SARS-CoV-2 detection using rapid testing approach; definition of the diagnostic strategy with the highest level of accuracy for a molecular and a rapid test for the detection of SARS-CoV-2; notification of immunodiagnostic assays suitable for technical triage/screening of SARS-CoV-2, thus reducing costs and increasing testing capacity for resource-limited settings; as well as informed decision on testing approaches with suitable efficiency and turn-around time.

- All the study protocols here described, especially for evaluation of assays, will be adapted to future comprehensive WHO guidances once published.

## Supporting information

**S1 File. Study information sheet.**
(PDF)

**S2 File. Informed consent sheet.**
(PDF)

**S3 File. Data collection tool.**
(PDF)

## Acknowledgments

We thank the various collaborative institutions for providing the required ressources for the development of the current study protocol.

## Author Contributions

**Conceptualization:** Joseph Fokam, Luna Colagrossi, Vittorio Colizzi, Nicaise Ndembi, Carlo-Federico Perno.

**Funding acquisition:** Joseph Fokam, Claudia Alteri, Anne-Marie Genevieve, Vittorio Colizzi, Nicaise Ndembi, Carlo-Federico Perno.

**Methodology:** Joseph Fokam, Claudia Alteri, Luna Colagrossi, Anne-Marie Genevieve, Désiré Takou, Vittorio Colizzi, Nicaise Ndembi, Carlo-Federico Perno.

**Project administration:** Alexis Ndjolo, Vittorio Colizzi.

**Resources:** Alexis Ndjolo.

**Software:** Claudia Alteri.

**Supervision:** Alexis Ndjolo, Nicaise Ndembi, Carlo-Federico Perno.

**Validation:** Joseph Fokam, Claudia Alteri, Luna Colagrossi, Anne-Marie Genevieve.

**Visualization:** Désiré Takou, Vittorio Colizzi, Carlo-Federico Perno.

**Writing – original draft:** Joseph Fokam, Claudia Alteri, Luna Colagrossi, Nicaise Ndembi.

**Writing – review & editing:** Joseph Fokam, Claudia Alteri, Luna Colagrossi, Anne-Marie Genevieve, Désiré Takou, Alexis Ndjolo, Vittorio Colizzi, Nicaise Ndembi, Carlo-Federico Perno.

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
