## [Decision Letter · Decision Letter 0]

16 Aug 2022

Diagnostic Performance of Molecular and Serological Tests of SARS- CoV-2 on well-Characterised Specimens from COVID-19 Individuals: the EDCTP "PERFECT-Study" (RIA2020EF-3000)

PONE-D-22-20470

Dear Dr. Fokam,

We’re pleased to inform you that your "study protocol" manuscript has been judged scientifically suitable for publication and will be formally accepted for publication once it meets all outstanding technical requirements.

*You are required to include the word “Protocol” in your Title.*

Kind regards,

Massimiliano Galdiero, M.D., Ph.D.

Academic Editor

PLOS ONE

Journal Requirements:

2. Thank you for submitting the above manuscript to PLOS ONE. During our internal evaluation of the manuscript, we found significant text overlap between your submission and previous work in the [introduction, conclusion, etc.].

Please revise the manuscript to rephrase the duplicated text, cite your sources, and provide details as to how the current manuscript advances on previous work. Please note that further consideration is dependent on the submission of a manuscript that addresses these concerns about the overlap in text with published work.

[If the overlap is with the authors’ own works: Moreover, upon submission, authors must confirm that the manuscript, or any related manuscript, is not currently under consideration or accepted elsewhere. If related work has been submitted to PLOS ONE or elsewhere, authors must include a copy with the submitted article. Reviewers will be asked to comment on the overlap between related submissions (http://journals.plos.org/plosone/s/submission-guidelines#loc-related-manuscripts).]

We will carefully review your manuscript upon resubmission and further consideration of the manuscript is dependent on the text overlap being addressed in full. Please ensure that your revision is thorough as failure to address the concerns to our satisfaction may result in your submission not being considered further

Reviewers' comments:

Reviewer's Responses to Questions

**Comments to the Author**

1. Does the manuscript provide a valid rationale for the proposed study, with clearly identified and justified research questions?

Reviewer #1: Yes

Reviewer #2: Yes

2. Is the protocol technically sound and planned in a manner that will lead to a meaningful outcome and allow testing the stated hypotheses?

Reviewer #1: Yes

Reviewer #2: Yes

3. Is the methodology feasible and described in sufficient detail to allow the work to be replicable?

Reviewer #1: Yes

Reviewer #2: Yes

4. Have the authors described where all data underlying the findings will be made available when the study is complete?

Reviewer #1: Yes

Reviewer #2: Yes

5. Is the manuscript presented in an intelligible fashion and written in standard English?

Reviewer #1: Yes

Reviewer #2: Yes

6. Review Comments to the Author

You may also provide optional suggestions and comments to authors that they might find helpful in planning their study.

Reviewer #1: The manuscript is very well written and shows an innovative tool to monitoring diagnostic performance of molecular and Serological Tests of SARS- CoV-2. The manuscript is clearly written, the set of methods is adequate to the goals formulated.

The topic is interesting, due to a lack of literature dealing with this matter and data support the interpretation of the discussion.

I suggest accepting the paper in his form

Reviewer #2: The authors present a research proposal aimed to implement measures to minimize risks for COVID-19 in Cameroon. The multidisciplinary approach aims to improve the diagnostic performances for COVID 19 by better integrating methodologies, and infrastructures already available to enhance capacities in Cameroon for detecting the infection.

Authors also plan to validate point-of-care SARSCoV-2 molecular assays and to implement SARS-CoV-2 diagnosis with ddPCR-based assays , to validate serological assays to identify COVID-19- exposed persons and follow-up of convalescents.

The prospective, observational study of 24 monthswill be performed in Cameroon by collecting 1,536 nasopharyngeal swabs and sera samples of COVID-19 contacts. Molecular testing will be performed on conventional real-time qPCR, point-of-care GeneXpert, antigen-tests and digital droplet PCR (ddPCR); testing will be performed using ELISA, and antibody-based kits. Sensitivity, specificity, positive- and negative-predictive values will be evaluated.

The dta obtained will contribute to generate the technical and clinical environment required for a better detection of Sars-CoV2- in Cameroon and in Africa in general

The project will achieve several important results which include the technology transfer for conventional and point-of-care molecular assays, the improvement of the turn-around-time for diagnosing COVID-19 infection with low-cost assays, allowing timely treatment. Finally the project will contribute to assess the epidemiology of COVID-19 and the circulating-variants in Cameroon

This project is interesting and timely and will achieve important result for the control pf COVID19 pandemic in Cameroon and in Africa

7. PLOS authors have the option to publish the peer review history of their article (what does this mean?). If published, this will include your full peer review and any attached files.

Reviewer #1: No

Reviewer #2: No

---

## [Editor Report · Acceptance letter]

12 Sep 2022

PONE-D-22-20470 

Diagnostic performance of molecular and serological tests of SARS-CoV-2 on well-characterised specimens from COVID-19 individuals : the EDCTP "PERFECT-study" protocol (RIA2020EF-3000) 

Dear Dr. Fokam:

I'm pleased to inform you that your manuscript has been deemed suitable for publication in PLOS ONE. Congratulations! Your manuscript is now with our production department. 

Kind regards, 

on behalf of

Prof. Massimiliano Galdiero 

Academic Editor

PLOS ONE